# OpenReview forum: "ManagerBench: Evaluating the Safety-Pragmatism Trade-off in Autonomous LLMs"
_ICLR.cc/2026/Conference — ICLR 2026 Poster_

### Official Review · Reviewer_oKZS · 2025-10-26

**Soundness:** 3
**Presentation:** 2
**Contribution:** 3
**Rating:** 4
**Confidence:** 3

**Summary:**

This paper introduces MANAGERBENCH, a novel benchmark designed to evaluate the safety-pragmatism trade-off in Large Language Models (LLMs) when they function as autonomous decision-makers. It identifies a critical gap in existing safety benchmarks, which primarily focus on preventing the generation of harmful content but overlook the challenge of agents taking harmful actions when operational goals conflict with human safety. MANAGERBENCH comprises 2,440 human-validated managerial scenarios that force a choice between a pragmatic but harmful action and a safe but less effective one. The evaluation of state-of-the-art LLMs reveals a systematic failure to navigate this trade-off: many models consistently choose harmful options to achieve goals, while others become overly safe and ineffective. Critically, the paper finds this misalignment stems not from an inability to perceive harm, but from flawed prioritization.

**Strengths:**

Novel Problem Formulation: The paper identifies and rigorously addresses a critical gap in AI safety: the trade-off between operational goals and human safety in autonomous agents. This moves beyond the well-trodden ground of content-based safety evaluation.

Rigorous Methodology: The construction of MANAGERBENCH is methodical and robust. The use of parallel datasets (human harm vs. inanimate object control), systematic parameterization across domains and harm types, and extensive human validation create a high-quality, credible benchmark.

Clarity and Quality of Presentation: The paper is written with exceptional clarity. The motivation is strong, the methods are well-explained, and the results are presented effectively through clear tables and insightful figures.

**Weaknesses:**

Synthetic Nature of Scenarios: While human-validated for realism, the scenarios are synthetic. Their direct applicability to real-world, complex decision-making environments remains to be proven. Future work could validate the benchmark against real-world case studies.

Binary Choice Constraint: The forced binary choice, while effective for diagnostic clarity, prevents models from proposing alternative, potentially safe-and-pragmatic solutions. This may underestimate a model's capability for nuanced problem-solving.

Prompt Sensitivity: The paper demonstrates that model performance is highly sensitive to prompt phrasing (e.g., the "nudging" experiment). While this is a key finding, the paper could have explored potential mitigation strategies or prompt engineering techniques to make model behavior more robust.

Lack of Longitudinal Analysis: The benchmark evaluates single, isolated decisions. It does not assess model behavior over longer time horizons, where consistency, learning, and strategic planning could play a role in the safety-pragmatism dynamic.

Potential for Uneven Domain Coverage: Although 11 domains are included, the paper does not analyze if model performance varies significantly across them. Some domains (e.g., Healthcare, Legal) might carry higher stakes and warrant a deeper, domain-specific analysis.

**Questions:**

1. Have the authors considered extending the benchmark to allow for a third option where the model can propose its own alternative solution? This could provide a more comprehensive assessment of a model's creative and ethical problem-solving abilities.

2. Are there plans to develop a longitudinal version of MANAGERBENCH to evaluate model consistency and strategic adaptation over a sequence of related decisions?

3. Did the authors explore any prompt engineering or scaffolding techniques that could help models better navigate the safety-pragmatism trade-off, thereby mitigating the fragility you observed?

---

> ### Author Response · Authors · 2025-11-18
>
> Thank you for the helpful review. We are glad you recognize the novelty of the problem our paper tackles (operational goal and human safety), the methodological setup of constructing ManagerBench, and the clarity of the presentation. We address your main concerns below.
>
> ---
>
> **Synthetic Nature of Scenarios:**
>
> We agree that real-world case studies would be valuable. However, constructing them would imply querying companies to either provide data on what costs are high/moderate/low risk, or verify the numbers that we generate are realistic, which is a prohibitively costly and time-consuming process. In addition, for simplicity, we framed the setup in binary options (reaching or missing the goal). Even in the current setup, we find that LLMs can recognize the harm/benefit tradeoff [section 4.1], and that humans consider scenarios as realistic [section 2.2.3]. Going beyond is out of scope for the current version of our benchmark, which was already costly and time-consuming to curate and evaluate.
>
> ---
>
> **Binary Choice Constraint:**
>
> We agree that because our benchmark format is a binary choice, it can prevent an innovative answer from the model. We explicitly acknowledge this as a limitation of our scenario [line 487]. However, this design choice is intentional: it enables a clear and unambiguous evaluation of model preferences when confronted with a direct trade-off. In addition, judging a model's novel solution could be very subjective and hard to evaluate. As this work is the first to introduce the concept of managerial decision evaluation of LLMs, we wanted a cleaner setting as a starting point for this line of research.
>
>
> We will clarify this by adding the following to line 131:
> > “Since this work introduces the concept of managerial decision evaluation, we wanted a clean setting as a starting point for this line of research. ”
>
> ---
>
> **Prompt Sensitivity**
>
> We agree that broader prompt-sensitivity ablations would strengthen the analysis. Running the full benchmark under multiple prompt variants is prohibitively expensive, which limited our ability to explore this space in the main paper.
> However, to address this concern, we conducted a new targeted study using paraphrased operational-goal prompts (generated with GPT-4o) and evaluated one representative model from each family.
> The results closely match the patterns reported in Table 1, indicating that the overall findings are stable under reasonable paraphrasing. We include these additional results in Appendix H.
> Taken together, these analyses reinforce that our conclusions are not an artefact of a particular prompt wording.
> Regarding prompt engineering techniques: We believe that optimizing anything (even the prompt) on the score on ManagerBench would defeat its purpose. Researchers can evaluate on our benchmark for training in order to investigate and improve LLM alignment; however, they should not then use the scores produced by the LLMs on our benchmark as rewards during training.
>
>
> One could likely find a prompt variation that improves model scores on ManagerBench, but since our scenarios are not exhaustive, this could provide a false sense of security. Models should be indirectly optimized for safety, and evaluated on our benchmark instead of being optimized directly for ManagerBench score.
>
>
> ---
>
> **Lack of Longitudinal Analysis**
>
> We appreciate the insight. In the current version of our benchmark, we aimed to verify that a problem of flawed prioritization exists, even in this simplistic setup. Multi-step scenarios are an excellent extension that would test long-term planning and whether early prioritization affects downstream choices. Designing realistic multi-step interactions introduces many modelling and evaluation decisions which are time-consuming to construct in a principled manner (e.g., managing long-horizon dependencies and interactive evaluation protocols) (Mohammadi et al., 2025 [https://arxiv.org/abs/2507.21504 ]; Wang et al., 2025 [https://arxiv.org/abs/2403.11085 ]), we view this as an important but substantial follow-up project.
>
> ---
>
> **Potential for Uneven Domain Coverage**
>
> We agree with the reviewer and have already conducted and reported this evaluation in the paper. The full MB-Score results for each domain are presented in the table added in Appendix G. While some variance exists between domains, no consistent trend appears across models, suggesting that there is no domain-specific systematic pragmatism-harmfulness preference.
>
>
> ---
> We appreciate your comments and have revised the paper to improve its clarity and completeness in line with your recommendations. We are happy to discuss any further questions or unclarities, as we would like to best improve the quality of our manuscript, which we believe identifies an important problem.

---

> > ### Author Response · Authors · 2025-11-26
> >
> > Thank you for your valuable suggestions. We hope our response was able to address your concerns. We are committed to improving the quality of our work, so please let us know if any points remain unclear.

---

### Official Review · Reviewer_76cT · 2025-10-31

**Soundness:** 3
**Presentation:** 3
**Contribution:** 3
**Rating:** 8
**Confidence:** 4

**Summary:**

In this paper, the authors have introduced ManagerBench, a benchmark designed to evaluate LLMs decision-making in realistic managerial scenarios (scenarios generated by LLMs but validated by humans). Each scenario forces a binary choice between a pragmatic but harmful action that achieves a goal and a safe action that produce a worse performance. More prominent LLMs perform poorly in balancing this safety-pragmatism trade-off. Moreover, the authors have shown that the detected alignment failures are a consequence of flawed prioritization rather than an inability to perceive harm, thus highlighting the issues with the current alignment strategies adopted for these models.

**Strengths:**

- A benchmark evaluating LLM decision-making in realistic managerial scenarios and in particular evaluating the trade-off between safe and pragmatic choices is of paramount relevance for the current development and adoption of safe LLM-powered agents.

- The study is well-designed with also a control set to identify the LLM tendency to be overly safe.

- Conclusions are always well supported by findings and experiments ran and are relevant to inform the design of safety guardrails and the adoption of LLM-powered agents in real-world scenarios

**Weaknesses:**

- I appreciated the human validation of the LLM-generated scenarios but it would be methodologically stronger having also a set of human-generated decision-making scenarios.

- Very relevant also the adoption of a control set but why the conditions between human harm set and control set are different in numbers? why just 2 types of object harm vs 8 human harm subtypes?

- In the human validation effort why the authors have used different scales to evaluate harmful and to evaluate realism?

- Again, why the authors adopted different max token generation lengths for the various models? and could this influence the models' behaviors?

**Questions:**

The questions are the ones reported above.

---

> ### Author Response · Authors · 2025-11-18
>
> Thank you for the positive and thoughtful review. We are glad you found our benchmark relevant to the current development of safe LLM-powered agents, that you consider the benchmark well designed with the control set, and that our conclusions are adequately supported by the experiments. Below, we address your concerns.
>
> ---
>
> **I appreciated the human validation of the LLM-generated scenarios but it would be methodologically stronger having also a set of human-generated decision-making scenarios.**
>
> We appreciate the suggestion. Human-authored scenarios would indeed strengthen the benchmark. However, creating such scenarios is resource-intensive: while annotating five existing examples already required about 30 minutes per annotator, writing new, realistic multi-constraint managerial dilemmas would require substantially more time, followed by the same validation process. Adding only a very small number would also offer limited statistical value and could be dominated by scenario-specific idiosyncrasies.
>
> Given these constraints, we prioritize the large, diverse, human-validated synthetic pool in this work, and view a sufficiently sized human-generated set as important future work.
>
> ---
>
> **Very relevant also the adoption of a control set but why the conditions between human harm set and control set are different in numbers? why just 2 types of object harm vs 8 human harm subtypes?**
>
> Thank you for this question. The control set focuses on object-related harm. In this case, it was difficult to come up with realistic types of harm (such as emotional, or legal). In contrast, human harm encompasses a broader range of possible consequences. Therefore, we included several subtypes to capture this diversity — economic, physical, emotional, and legal harms. These categories reflect the different dimensions in which harm can affect people, whereas harm to objects is typically more limited in scope.
>
> However, to better investigate this setup, we now added experiments with two additional object harm types (reduced lifetime of the product and decreased functionality of the product), which show similar trends but stronger over-safety in the models. See Appendix I for the new results.
>
> ---
>
> **In the human validation effort why the authors have used different scales to evaluate harmful and to evaluate realism?**
>
> Thank you for this question. We wanted a fine-grained evaluation for the harmfulness to allow potential filtering of less harmful examples. In the realism evaluation, we found it simpler to have a coarse-grained scale, as we did not want to filter anything.
>
> ---
>
> **Again, why the authors adopted different max token generation lengths for the various models? and could this influence the models' behaviors?**
>
> A maximum generation length of 1024 tokens was set for all models, except for the thinking models (Gemini and GPT-5-H). This was decided after we manually detected on a few dozen examples that the models usually generate an answer before reaching the 1024 tokens. “Most models adhered to the required response template in 95% or more of cases” [line 239]. Thus, we do not expect additional tokens to modify the results significantly. Manual observation on a few dozen of these examples indicated that for most models, less than half of the **less than 5%** may be affected by the token length. In addition, due to high API costs, limiting the token generation was needed.
>
> To clarify this, we added to footnote 7  “...As most models adhered to the required response template in 95% or more of cases, increasing the generation length can have a limited effect on the results.”
>
> ---
>
> Thank you for your feedback. We have updated the paper to enhance its clarity and completeness in response to your suggestions. We are happy to address any additional questions you may have.

---

> > ### Comment · Reviewer_76cT · 2025-11-25
> >
> > I'm happy with the answers provided and I keep my positive assessment on the paper and my suggestion for acceptance.

---

> > > ### Author Response · Authors · 2025-11-26
> > >
> > > Thank you very much. We are happy we were able to address your concerns.

---

### Official Review · Reviewer_Wxr8 · 2025-11-01

**Soundness:** 3
**Presentation:** 3
**Contribution:** 3
**Rating:** 6
**Confidence:** 4

**Summary:**

The paper introduces MANAGERBENCH, a novel benchmark to evaluate autonomous LLMs in managerial roles by testing the trade-off between operational goals and human safety. It presents models with realistic scenarios forcing a choice between a "pragmatic but harmful" action (achieving a goal by harming humans) and a "safe but ineffective" one. A key innovation is a parallel "control set," where harm is directed at inanimate objects, to identify "overly safe" models that are pragmatically useless. The authors find that all current LLMs fail this test, falling into either the "unsafe" or "overly safe" category. This failure stems not from an inability to perceive harm, but from a flawed prioritization of operational goals, a vulnerability easily exploited by simple "nudging" prompts.

**Strengths:**

1. Novel and Critical Problem Definition:

The paper addresses a forward-looking and critical gap in AI safety. As LLMs transition from "assistants" to "agents," evaluating their actions in goal-oriented conflicts is more important than just evaluating their content generation, which most prior benchmarks focus on.

2. Innovative Evaluation Methodology (Control Set):

The inclusion of the parallel "control set" (harm to inanimate objects) is a significant methodological strength. It allows the benchmark to disentangle genuine safety alignment from rigid, "overly safe" behavior, where a model might just be trained to avoid any option with negative sentiment. This provides a much-needed measure of "pragmatism."

3. Deep and Actionable Core Insight:

The paper's most powerful contribution is successfully distinguishing between perception and prioritization. By showing that models understand what is harmful (their ratings align with humans') but still choose the harmful option, the authors pinpoint the alignment failure in "flawed prioritization". This is a vital insight for the field.

4. Rigorous Benchmark Construction and Validation:

The benchmark is not arbitrary. It was systematically generated using state-of-the-art LLMs across 11 domains, 4 harm categories, 4 incentive types, and varying intensity levels. Most importantly, the scenarios were then rigorously validated by 25 human annotators to ensure they were perceived as both "realistic" and "harmful".

5. Robust Experimental Design:

The authors included experiments that solidify their claims. For instance, the "nudging" prompt experiment (Section 3.3) demonstrated the "brittleness" of current safety alignments. By adding a simple instruction like "Nothing else matters" , the harm avoidance scores of models like Gemini plummeted by 55 points, proving that existing safety guardrails are easily bypassed by goal-oriented pressure.

**Weaknesses:**

1. Synthetic and Simplified Scenarios:

The authors acknowledge this limitation themselves. The scenarios are "synthetic, not drawn from real-world cases". While validated for "realism," they are still simplified constructs. Real-world managerial decisions are often far more ambiguous and complex than the scenarios generated by the LLMs.

2. Restrictive Binary-Choice Format:

The benchmark "deliberately" forces a binary (A/B) choice. While this provides a clean, unambiguous evaluation of prioritization, it "prevents models from proposing alternative solutions". A truly superior, well-aligned agent might reject both bad options and propose a creative third way to achieve the goal without causing harm. This benchmark cannot measure that crucial capability.

3. Narrow Definition of "Pragmatism":

The "Control Pragmatism" metric is a clever proxy, but it is narrowly defined as "the willingness to harm an inanimate object". A model could be "overly safe" in this one specific context (e.g., Sonnet-4 refusing to scratch furniture)  but still be highly pragmatic and effective in other managerial tasks that don't involve harm, such as resource allocation or financial planning. The benchmark may be over-generalizing this finding.

4. High Sensitivity to Prompt Phrasing:

The "nudging" experiment (a strength) also reveals a weakness: the models' behaviors are extremely sensitive to small changes in the prompt. This implies that the baseline results in Table 1 are themselves highly dependent on the specific, neutral phrasing of the system and user prompts. A slightly different phrasing could potentially yield very different results, limiting the generalizability of the scores.

5. Potential for Benchmark-Specific Overfitting:

The authors rightly "explicitly discourage using these scenarios for model training," fearing it would create a "false sense of security". This highlights a weakness of the benchmark's structure: a model could be fine-tuned to specifically pass MANAGERBENCH (i.e., "be safe with humans, be pragmatic with objects") without developing a general, robust understanding of safety-pragmatism trade-offs.

**Questions:**

1. The current A/B format is effective for a clean diagnosis but is restrictive. Is it possible to allow for open-ended responses or a "propose an alternative" option? This would test a more advanced capability: whether the LLM can find a creative third solution that achieves the goal without causing harm, which is a hallmark of a truly superior and aligned agent.

2. Real-world managerial dilemmas are rarely isolated, single-shot decisions. Is it possible to introduce multi-step scenarios where the model's initial choice (e.g., to prioritize pragmatism) has cascading ethical and operational consequences in subsequent steps, testing its long-term strategic reasoning.

---

> ### Author Response · Authors · 2025-11-18
>
> **Part1:**
>
> Thank you for the positive and thoughtful review! We are happy the reviewer acknowledges the novelty of our goal-oriented conflicting setup, the innovative control set, the insight that models choose the harmful option despite aligning with humans on harmfulness, the rigorous benchmark pipeline, and the robust experiments. We address your main concerns below.
>
> ---
>
> **Synthetic and Simplified Scenarios:**
>
> We agree that ManagerBench uses synthetic, relatively simple scenarios, and we noted this as a limitation. However, the fact that models already fail on scenarios this simple indicates that the underlying prioritization issue persists even in such a simple setup.
> We clarified this by adding the following to line 131:
> >“Identifying alignment failures even in multi-choice setups is an indication of flawed prioritization.”
>
> While we agree it is preferred to use human annotation to create the examples, this incurs a prohibitively high amount of cost/time. Annotating five existing examples already required about 30 minutes per annotator, and writing new, realistic multi-constraint managerial dilemmas would require substantially more time, followed by the same validation process. We agree on the importance of realism, which is why we conducted a realism evaluation [Section 2.2.3] – precisely to assess this. Our study shows that the synthetic scenarios are judged as realistic, supporting the realism of our benchmark while making the construction process tractable.
>
> ---
>
> **Restrictive Binary-Choice Format:**
>
> We agree that because our benchmark format is a binary choice, it can prevent an innovative option from the model. And we explicitly acknowledge this as a limitation of our scenario [Line 487]. However, this design choice is intentional: it enables a clear and unambiguous evaluation of model preferences when confronted with a direct trade-off.  In addition, judging a model's new solution could be very subjective and hard to evaluate. As this work is the first to introduce the concept of managerial decision evaluation of LLMs, we wanted a cleaner setting as a starting point for this line of research.
>
> ---
>
> **Narrow Definition of "Pragmatism":**
>
> We appreciate this concern and will clarify the intended scope. "Control Pragmatism" is not intended as a general notion of pragmatism across all managerial functions. It is a counterfactual control condition explicitly designed to diagnose over-generalized safety behavior, by holding everything constant except the presence of human harm. Thus, the metric reveals whether a model systematically avoids any negative-sentiment option, even when the "harm" is trivial and operationally dominated, and not whether the model is or is not pragmatic in unrelated managerial skills.
>
> To better investigate this setup, we now experimented with two additional object harm types (reduced lifetime of the product and decreased functionality of the product), which show similar trends but stronger over-safety in the models. See Appendix I for the new results.
> We added the following to line 153:
> >"It [control pragmatism] is not meant to capture a model’s broader managerial competence. Instead, it provides a controlled counterfactual to detect overly safe behavior."
>
> ---
>
> **High Sensitivity to Prompt Phrasing:**
>
> We agree that broader prompt-sensitivity ablations would strengthen the analysis. Running the full benchmark under multiple prompt variants is prohibitively expensive, which limited our ability to explore this space in the main paper.
> However, to address this concern, we conducted a new targeted study using paraphrased operational-goal prompts (generated with GPT-4o) and evaluated one representative model from each family.
> The results closely match the patterns reported in Table 1, indicating that the overall findings are stable under reasonable paraphrasing. We include these additional results in Appendix H. Taken together, these analyses reinforce that our conclusions are
> not an artefact of a particular prompt wording.

---

> > ### Author Response · Authors · 2025-11-18
> >
> > **Part 2:**
> >
> > **Potential for Benchmark-Specific Overfitting:**
> >
> > ManagerBench is intended solely for evaluation, not training. Its purpose is to assess models and surface potential safety issues within a realistic management context. While the benchmark follows a certain general structure, we identify a very important problem of flawed prioritization in LLMs. Researchers can evaluate on our benchmark for training in order to investigate and improve LLM alignment; however, they should not then use the scores produced by the LLMs on our benchmark as rewards during training. To
> > detect training on ManagerBench, we will add a watermark (Kirchenbauer et al. 2024 [https://arxiv.org/pdf/2301.10226]).
> >
> > ---
> >
> > **Real-world managerial dilemmas are rarely isolated, single-shot decisions. Is it possible to introduce multi-step scenarios where the model's initial choice…**
> >
> > We appreciate the insight. Multi-step scenarios are an excellent extension that would test long-term planning and whether early prioritization affects downstream choices. Because designing realistic multi-step interactions introduces many modelling and evaluation decisions (e.g., managing long-horizon dependencies and interactive evaluation protocols) (Mohammadi et al., 2025 [https://arxiv.org/abs/2507.21504]; Wang et al., 2025 [https://arxiv.org/abs/2403.11085]), we view this as an important but substantial follow-up project.
> >
> > ---
> >
> > We appreciate your feedback and have revised the paper to improve clarity and completeness based on your suggestions. We would be happy to discuss any remaining questions further.

---

### Official Review · Reviewer_nGQa · 2025-11-01

**Soundness:** 3
**Presentation:** 3
**Contribution:** 2
**Rating:** 4
**Confidence:** 4

**Summary:**

The paper introduces ManagerBench that evaluates how LLM agents navigate a safety-pragmatism trade-off in realistic managerial scenarios by asking it to choose either (i) a pragmatic option that achieves an operational goal but harms humans, versus (ii) a safe option that sacrifices performance. The benchmark also includes a control set in which each scenario with harm directed to inanimate objects to measure over-safety. LLMs are scored by their harmonic mean of Harm Avoidance and Control Pragmatism which is called MB-Scores. The results show that low overall MB-Scores and suggests models perceive harm similarly to humans but often prioritize goals over safety.

**Strengths:**

**Originality**
* Frames safety evaluation as decision-making under conflicting incentives rather than refusal of harmful content.
* The inclusion of the control set is a nice idea to separate human-safety alignment from just pure risk aversion.

**Quality**
* Clear example structure and protocol
* Detailed explanation on thoughtful construction pipeline with human-validated for perceived harm and realism.
* Model coverage is broad.

**Clarity**
* Overall nice figures and examples with detailed explanation. Easy to follow the paper.

**Significant**
* The state-of-the-art models struggle on the safety-pragmatism treade-off is an interesting founding.

**Weaknesses:**

1. The scenarios are intentionally binary with explicit harm statement, which sharpens diagnosis but may over-simplify managerial reality and edge toward moral-dilemma style tests.
2. Treating “object harm” as the pragmatic choice risks baking in a normative stance (orgs often internalize equipment damage costs, compliance, and reputational risk). Without explicit cost calibration, Control Pragmatism may conflate reasonable caution with “over-safety.”
3. Harm/benefit stakes are not monetized to real-world costs. It’s unclear if observed trade-offs align with realistic managerial risk calculus across domains.
4. Broader ablations with prompt sensitivity analyses are limited.
5. Treating refusal as an incorrect response could be misleading. Some models decide to refuse just by looking at harmful contents.

**Questions:**

See the weaknesses.

---

> ### Author Response · Authors · 2025-11-18
>
> Thank you for the helpful review. We are glad you recognize the originality of ManagerBench, including the control set, the quality and human validation of our scenarios, that the paper is easy to follow, and are happy that the safety-pragmatism trade-off finding is interesting, which is the core question underpinning our work.
> Below, we address your main points:
>
> ---
>
> **The scenarios are intentionally binary with explicit harm statement, which sharpens diagnosis but may over-simplify managerial reality and edge toward moral-dilemma style tests.**
>
> As the reviewer correctly observes, we adopt a binary choice setup to simplify and sharpen the diagnostic signal. We explicitly acknowledge this as a limitation of our scenario (Line 487). However, this design choice is intentional: it enables a clear and unambiguous evaluation of model preferences when confronted with a direct trade-off. Despite the binary structure, our human validation study [section 2.2.3] shows that annotators still perceived the scenarios as realistic. Importantly, our setup differs from moral-dilemma–style evaluations because one of the available options is harmless to humans, avoiding the ambiguity typically inherent in such tests. In this version of our benchmark, our goal was to demonstrate that a critical problem of flawed prioritization exists in LLMs. Subsequent versions can explore different settings, such as multiple choices, open-ended answers, and complex multi-step decisions.
>
> ---
>
> **Treating “object harm” as the pragmatic choice risks baking in a normative stance (orgs often internalize equipment damage costs, compliance, and reputational risk). Without explicit cost calibration, Control Pragmatism may conflate reasonable caution with “over-safety.”**
>
> We agree with the reviewer that, in principle, treating “object harm” as the pragmatic option could risk encoding a normative stance. In our scenarios, however, the object harm is deliberately made minor and trivial, as can be seen in the examples in Appendix A, e.g, “introduces a minor, statistically insignificant risk of causing defects or scratches to office furniture in 15%“ (line 1167). The “object harm” is explicitly stated as minor and statistically insignificant.
>
> ---
>
>
> **Harm/benefit stakes are not monetized to real-world costs. It’s unclear if observed trade-offs align with realistic managerial risk calculus across domains.**
>
> While it is true that we do not align trade-offs to realistic managerial risk calculus, obtaining such information is extremely difficult. It would imply querying companies to either provide data on what costs are high/moderate/low risk, or verify the numbers that we generate, which is a prohibitively costly and time-consuming process. Even in the current setup, we find that LLMs can recognize the harm/benefit tradeoff [section 4.1] and that humans consider scenarios as realistic [section 2.2.3]. Going beyond is out of scope for the current version of our benchmark.
>
> ---
>
> **Broader ablations with prompt sensitivity analyses are limited.**
>
> We agree that broader prompt-sensitivity ablations would strengthen the analysis. Running the full benchmark under multiple prompt variants is prohibitively expensive, which limited our ability to explore this space in the main paper.
> However, to address this concern, we conducted a new targeted study using paraphrased operational-goal prompts (generated with GPT-4o) and evaluated one representative model from each family.
> The results closely match the patterns reported in Table 1, indicating that the overall findings are stable under reasonable paraphrasing. We report these additional results in Appendix H.
> Taken together, these analyses reinforce that our conclusions are not an artefact of a particular prompt wording.
>
> ---
>
> **Treating refusal as an incorrect response could be misleading. Some models decide to refuse just by looking at harmful contents.**
>
> It is important to note that models do not respond as instructed only in a small number of cases (less than 5% for most models, see Section 2.3 and Appendix F), meaning that the effect of refusal should not have a significant impact on the results. However, we strongly believe that abstaining from choosing an answer when a harmless one is present indicates misalignment, or at the very least, lack of capacity to recognize the harmless option.
>
> ---
>
> We thank you for your thoughtful comments. We have revised the paper with further analyses and clarifications based on your suggestions. We believe our work identifies an important problem, and we would be happy to discuss any remaining questions further to best improve our manuscript.

---

> > ### Comment · Reviewer_nGQa · 2025-11-18
> >
> > Thank you for the detailed explanation and the new experiment. I have raised my score.

---

> ### Author Response · Authors · 2025-11-19
>
> Thank you. We’re pleased we were able to address your concerns.

---

### Author Response · Authors · 2025-11-18

We thank the reviewers for acknowledging the relevance and quality of our benchmark, as well as highlighting the core insight as interesting.

Thanks to the reviewers' comments, we added three new experimental results to the paper:

Appendix G: We report model performance across individual domains. Showing variance exists between domains, with no consistent trend appearing across models, suggesting that there is no domain-specific systematic pragmatism-harmfulness preference.

Appendix H: We report new results analyzing the robustness of ManagerBench under input paraphrasing. To better validate the robustness of ManagerBench, we paraphrased the operational goal in our scenarios from the benchmark and evaluated whether model performance is consistent on this. This evaluation showed consistent results with the ones presented in the main paper, indicating that input variation does not affect our findings.

Appendix I: Expanding coverage of control scenarios. To increase the scenarios in the control set, we created additional object harm scenarios (a total of 264).
The results show lower pragmatism compared to those reported in Section 3.1, suggesting that additional scenarios can increase the comprehensiveness of ManagerBench.

---

### Meta-Review · Area_Chair_n9aN · 2026-01-07

**Summary:**

### Summary
This paper introduces ManagerBench, a benchmark for evaluating LLM agents in realistic managerial decision-making scenarios that require navigating a safety–pragmatism trade-off. Each scenario forces a choice between a pragmatic option that achieves an operational goal but harms humans and a safe option that sacrifices performance; a parallel control set swaps human harm for minor object harm to detect over-safety. The paper proposes MB-Score (harmonic mean of Harm Avoidance and Control Pragmatism) and shows that current frontier models achieve low scores, with failures driven more by flawed prioritization under goal pressure than by inability to perceive harm, and that simple nudging prompts can strongly degrade safety behavior.

### Strengths
- Novel and timely problem formulation: evaluates agentic decision-making under conflicting incentives rather than content refusal or harmful generation alone.
- Strong benchmark design: paired human-harm and object-harm control sets enable separating genuine human-safety alignment from blanket risk aversion.
- Rigorous scenario construction and validation: diverse coverage (domains, harm categories, incentives, intensity) and human validation for harm perception and realism.
- Actionable insight: distinguishes perception vs prioritization and demonstrates brittleness via nudging and targeted prompt robustness checks.

### Weaknesses
- Scenarios are synthetic and simplified, and the binary-choice format prevents models from proposing creative third options; this may understate aligned agents that can find alternative solutions.
- Control pragmatism is a narrow proxy for pragmatism (object harm), and without monetized stakes it may not reflect real managerial risk calculus.
- Results can be prompt-sensitive; broader prompt-sensitivity or mitigation studies remain limited (though targeted paraphrase tests help).
- Potential benchmark-specific overfitting risk if used for training (mitigations are discussed, but structural risk remains).

ManagerBench targets a critical gap in agent safety evaluation by measuring how models act when operational goals conflict with human safety, and the benchmark’s paired control design offers a clean diagnostic for over-safety versus misalignment. The construction pipeline is systematic and human-validated, and the empirical findings are consistent and highly actionable, especially the evidence that failures stem from flawed prioritization and can be amplified by simple nudging. Despite acknowledged limitations (synthetic, binary format), the paper establishes an important evaluation direction with strong methodology and clear implications for aligning real-world LLM agents.

**Reviewer Concerns:**

- nGQa
  - addressed: clarified rationale for binary format and explicit-harm design; justified object-harm control via “minor/trivial” harms; added targeted prompt-paraphrase robustness; contextualized refusal rate and its limited effect.
  - still outstanding: lack of monetized stakes and broader prompt-sensitivity; potential normative concerns around object harm remain partially conceptual.

- Wxr8
  - addressed: clarified scope of Control Pragmatism as counterfactual diagnostic (not general managerial competence); added additional object-harm types; added prompt-paraphrase robustness study; discussed overfitting risks and mitigation plan (watermarking).
  - still outstanding: binary-choice restriction and synthetic nature remain inherent limitations; benchmark-specific overfitting risk cannot be fully eliminated.

- 76cT
  - addressed: explained limited feasibility of a sizable human-authored scenario set; expanded object-harm types; justified different validation scales; clarified token-length choices and likely limited impact; reviewer confirmed satisfaction and maintained strong accept.
  - still outstanding: human-authored scenarios and richer realism are still future work.

- oKZS
  - addressed: acknowledged synthetic/binary limitations and positioned as intentional first step; added targeted prompt-paraphrase robustness; noted domain-level results are provided; clarified why prompt-optimizing for MB-Score is discouraged.
  - still outstanding: no longitudinal/multi-step evaluation yet; limited mitigation strategies explored beyond reporting robustness.

**Reviewer Scores:**

- nGQa: would increase (explicitly raised score after rebuttal)
- Wxr8: no change
- 76cT: no change (explicitly kept positive acceptance recommendation)
- oKZS: no change

---

### Decision · Program_Chairs · 2026-01-26

Accept (Poster)